



# Validation of formaldehyde products from three satellite retrievals (OMI SAO, OMPS-NPP SAO, and OMI BIRA) in the marine atmosphere with four seasons of ATom aircraft observations

Jin Liao[1,2], Glenn M. Wolfe[1], Alex E. Kotsakis[1,2,*], Julie M. Nicely[1,3], Jason M. St. Clair[1,2], Thomas F. Hanisco[1], Gonzalo González Abad[4], Caroline R. Nowlan[5], Zolal Ayazpour[5,6], Isabelle De Smedt[7], Eric C. Apel[8], Rebecca S. Hornbrook[8]

[1]Atmospheric Chemistry and Dynamic Laboratory, NASA Goddard Space Flight Center, Greenbelt, MD, USA
[2]Goddard Earth Sciences Technology and Research (GESTAR II), University of Maryland, Baltimore County, MD, USA
[3]Earth Resources Technology (ERT) Inc., Laurel, MD, USA
[4]Earth System Science Interdisciplinary Center (ESSIC), University of Maryland, College Park, MD, USA
[5]Center for Astrophysics Harvard-Smithsonian, Cambridge, MA, USA
[6]Department of Civil, Structural and Environmental Engineering, University of Buffalo, Buffalo, NY, USA
[7]Royal Belgian Institute for Space Aeronomy (BIRA-IASB), Brussels, Belgium
[8]Atmospheric Chemistry Observations & Modeling Laboratory, National Center for Atmospheric Research (NCAR), Boulder, CO, USA
[*]Now at Earth System Science Interdisciplinary Center (ESSIC), University of Maryland, College Park, MD, USA

*Correspondence to*: Jin Liao (jin.liao@nasa.gov)

**Abstract.** Formaldehyde (HCHO) in the atmosphere is an intermediate product from the oxidation of methane and non-methane volatile organic compounds. In remote marine regions, HCHO variability is closely related to atmospheric oxidation capacity and modeled HCHO in these regions is usually added as a global satellite HCHO background. Thus, it is important to understand and validate the levels of satellite HCHO over the remote oceans. Here we intercompare three satellite retrievals of total HCHO columns (OMI-SAO (v004), OMPS-NPP SAO, and OMI BIRA) and validate them against in situ observations from the NASA Atmospheric Tomography Mission (ATom) mission. All retrievals are correlated with ATom integrated columns over remote oceans, with OMI SAO (v004) showing the best agreement. Three satellite HCHO retrievals and in situ ATom columns all generally captured the spatial and seasonal distributions of HCHO in the remote ocean atmosphere. Retrieval bias varies by latitude and season, but a persistent low bias is found in all products at high latitudes and the general low bias is most severe for the OMI BIRA product. Examination of retrieval components reveals slant column corrections have a larger impact on the retrievals over remote marine regions while AMFs play a smaller role. This study informs that the potential latitude-dependent biases in the retrievals require further investigation for improvement and should be considered when using marine HCHO satellite data, and vertical profiles from in situ instruments are crucial for validating satellite retrievals.



## 1 Introduction


Formaldehyde (HCHO) in the marine atmosphere is mainly produced from oxidation of methane. Non-methane volatile
organic compounds (VOCs) transported from continents and potentially VOCs emitted at the ocean surface (Guenther et al.,
1995; Novak and Bertram, 2020) may also contribute to the marine HCHO. Methane is the dominant precursor of HCHO in
the remote atmosphere and oxidation of methane by hydroxyl radical (OH) represents ~ 80% of the global HCHO source
(Fortems-Cheiney et al., 2012; Wolfe et al., 2019). Satellite HCHO columns have been used to estimate the levels of
atmospheric oxidant OH, which plays an important role in removing air pollutants and greenhouse gas methane (Wolfe et al.,
2019). HCHO in the clean remote ocean atmosphere is considered as HCHO tropospheric background due to the short
atmospheric lifetime of HCHO of a few hours and its source locations. The column abundance of HCHO ranges from
~$1\times10^{15}$ molec cm$^{-2}$ in the remote troposphere (Vigouroux et al., 2018; Zhu et al., 2020) to the order of $10^{16}$ molec cm$^{-2}$ over
continental regions (Zhu et al., 2016).

HCHO is one of the few VOCs that can be observed from space. Satellite HCHO observations have been obtained by Global
Ozone Monitoring Experiment (GOME) (1995-2011) (Chance et al., 2000; Thomas et al., 1998), the Scanning Imaging
Absorption SpectroMeter for Atmospheric ChartographY (SCIAMACHY) (2002–2012) (De Smedt et al., 2008), GOME-2
(2006–2021/2012–present/2018–present) (De Smedt et al., 2012), the Ozone Monitoring Instrument (OMI) (2004–present)
(De Smedt et al., 2015; González Abad et al., 2015), the Ozone Mapping and Profiler Suite (OMPS) on Suomi NPP (Li et
al., 2015; González Abad et al., 2016; Nowlan et al., 2023) and on NOAA-20 (2017–present) (Nowlan et al., 2023), and the
TROPOspheric Monitoring Instrument (Sentinel-5P/TROPOMI) (2017–present) (De Smedt et al., 2021, 2018).
Geostationary satellite instruments also retrieve HCHO, including the Geostationary Environment Monitoring Spectrometer
(GEMS) (Kim et al., 2020; Kwon et al., 2019) over East Asia (2020–present), Tropospheric Emissions: Monitoring of
Pollution (TEMPO) (Chance et al., 2019) over North America (2023–present) and the upcoming European Sentinel-4
mission (Gulde et al., 2017). Major retrieval algorithms for HCHO include those developed by the Smithsonian
Astrophysical Observatory (SAO), Belgian Institute for Space Aeronomy (BIRA), and NASA Goddard Space Flight Center
(GSFC). These algorithms have evolved over time.

Previous studies have validated satellite HCHO retrievals with airborne and ground-based in situ and remote sensing
instruments in different settings and contexts. Zhu et al. (2016) indirectly evaluated six retrievals from four sensors against
airborne observations in the isoprene-rich Southeast U.S. using a model as an intermediary, finding a low bias in the mean
by 20–51% for all retrievals. Zhu et al. (2020) extend this method to indirectly validate OMI SAO v003 data with in-situ
HCHO measurements from 12 aircraft campaigns over North America, East Asia, and the remote Pacific Ocean. They found
that the OMI SAO v003 product has negative biases (-44:5% to -21:7%) under high-HCHO conditions and high biases
(+66:1% to +112:1 %) under low-HCHO conditions (Zhu et al., 2020). De Smedt et al. (2021) validated TROPOMI and





OMI-BIRA HCHO against a Multi-axis differential optical absorption spectroscopy (MAX-DOAS) ground network, finding
that compared to the MAX-DOAS ground network, TROPOMI HCHO columns are biased low especially for high
concentrations and OMI-BIRA HCHO columns are biased high at low concentrations and biased low at high concentrations
(De Smedt et al., 2021). In validation using Fourier transform infrared (FTIR) data, TROPOMI HCHO columns were biased
high for low concentrations sites and biased low for high concentrations sites and the correlation between TROPOMI and
FTIR HCHO columns yields a slope of 0.64 and an intercept of $1.10 \times 10^{15}$ molecules cm$^{-2}$ (Vigouroux et al., 2020). OMPS
Suomi NPP and NOAA-20 HCHO columns generally have good agreement with NDACC FTIR observations at 24 sites. The
linear regression between OMPS-NPP and FTIR HCHO columns yields a slope of 0.82 and an intercept of $5.71 \times 10^{14}$
molecules cm$^{-2}$ and the linear regression between OMPS-NOAA20 and FTIR reveals a slope of 0.92 and an intercept of 6.76
$\times 10^{14}$ molecules cm$^{-2}$ (Kwon et al., 2023). OMPS-NPP and OMPS-NOAA20 HCHO columns are also biased high
compared to FTIR measurements for sites with low HCHO levels (Kwon et al., 2023).

Most validation efforts focus on continental regions, while comparatively few examine the remote marine atmosphere. No
previous validation of satellite HCHO over the remote oceans with airborne in situ measurement was performed before the
NASA ATom field campaigns (2016–2018). OMI SAO v003 retrieval has been compared to two seasons of ATom
observations over both Pacific and Atlantic Oceans (Wolfe et al., 2019) and over the clean Pacific Ocean (Zhu et al., 2020),
with HCHO columns ranging from $1 \times 10^{15}$ to $8 \times 10^{15}$ molecules cm$^{-2}$. The ground FTIR HCHO measurements at Mauna
Loa in the Pacific Ocean domain are about $1 \times 10^{15}$ molecules cm$^{-2}$ for the background atmosphere measurements
(Vigouroux et al., 2018).

The accuracy of model predicted HCHO over the Pacific Ocean affects the global HCHO background in satellite retrievals.
In satellite HCHO retrievals, differential HCHO slant columns are often derived using spectra measured over a reference
sector in the Pacific Ocean, and modeled HCHO columns over the reference sector are added back to account for the real
HCHO levels over the reference sector (De Smedt et al., 2018; Nowlan et al., 2023). The locations of the area in the Pacific
Ocean used as reference sectors vary among different retrievals (De Smedt et al., 2018; Nowlan et al., 2023). Modeled
HCHO levels over the remote Pacific Ocean also play a role in correcting some biases such as latitude-dependent biases in
slant columns (De Smedt et al., 2018; Nowlan et al., 2023). Consequently, validating satellite HCHO over the remote ocean
would aid in assessing the satellite's ability to capture background HCHO levels accurately and enhancing our understanding
of these baseline levels.

Here we present a systematic comparison of in situ HCHO columns from four seasons of ATom observations with three
commonly-used satellite retrievals. Study objectives include 1) quantify spatial and seasonal retrieval bias, 2) quantify
differences between retrievals, and 3) identify relative contributions of retrieval components to inter-retrieval differences and
overall bias.




## 2. Methods

### 2.1 ATom observations

The NASA ATom mission studied atmospheric composition from near pole-to-pole over the Pacific and Atlantic remote
oceans with frequent vertical profiling from above the sea surface (100 m) to 10-12 km altitude for four seasons during
2016-2018 (Thompson et al., 2022).

The primary source of in situ HCHO measurements for this study is the In Situ Airborne Formaldehyde (ISAF) instrument
(Cazorla et al., 2015). ISAF data are reported at 1 Hz with a 1σ precision of 30 pptv. Systematic uncertainty is estimated as
10% + 10 pptv based on pre- and post-mission calibration against compressed gas standards. ISAF measurements are not
available during the second half of ATom 4, thus we also use HCHO observations from the Trace Organic Gas Analyzer
(TOGA) instrument (Apel et al., 2003, 2015). The TOGA reporting period is 2 minutes, and reported HCHO accuracy is
40% ± 40 pptv. Brune et al. (2020) performed a comparison of ISAF and TOGA data for all four ATom deployments and
found mission-to-mission variability in measurement agreement, with relatively good agreement for ATom-4. Similarly, we
find that the two measurements agree well for this deployment (Figure S1, slope of 1.1). Due to the higher accuracy and
measurement frequency of ISAF than TOGA, ISAF HCHO measurements from ATom are used when available.

ATom in situ HCHO composite columns are derived from the ATom vertical profiles. Ascents and descents occur along
transits between locations and typically cover 200-450 km in horizontal distance (Wolfe et al., 2019). In situ HCHO
columns are calculated using the method described in Wolfe et al. (2019). Each profile is averaged to an altitude grid of 0 to
10 km with 200 m spacing. Few measurements above 10 km are excluded. The lowest (or highest) altitude measurements are
extrapolated to the surface 0 km or (10 km) using the average of the two lowest (or highest) altitude measurements of that
profile. Missing data in between are linearly interpolated. Columns are filtered to include only profiles with solar zenith
angle smaller than 80°, minimum altitude <= 600 m, maximum altitude >= 8 km, fraction of interpolated grids < 0.2, and
fraction of extrapolation data <0.25. Average gas profiles from OMI SAO HCHO retrievals are used to estimate the
contribution of HCHO above 10 km to the total HCHO column. The calculated fraction of HCHO above 10 km (relative to
the total column) is 0.045± 0.002. This value is used to scale up in situ HCHO columns for comparison with satellite
retrievals.





## 2.2 Satellite HCHO retrieval products

### 2.2.1 OMI SAO (v004)

OMI was launched in 2004 onboard the NASA Aura satellite. It has a native spatial resolution at nadir of $24 \times 13$ km$^2$ (Table
1) with daily global coverage at a local overpass time of 13:30. The Smithsonian Astrophysical Observatory (SAO) version
004 retrieval is the updated version of OMI SAO v003 (González Abad et al., 2015) and is identical to the OMPS-NPP SAO
retrieval (Nowlan et al., 2023). The algorithm involves two main steps: 1) Following line shape and spectral calibration,
spectral fitting at 328.5-356.5 nm range for individual ground pixel is applied and a reference spectrum from a clean region
over the Pacific Ocean is used with the measured spectrum to derive the differential slant column ($\Delta$SCD), and 2) converting
the resultant $\Delta$SCD to vertical column density (VCD) using slant column corrections and the air mass factor (AMF). The
HCHO absorption cross section used in OMI SAO 004 is from Chance and Orphal (2011) at 300 K (Table 1). The location
of the reference spectrum is over the clean Pacific Ocean but varies slightly day-to-day due to orbit overpass location. The
OMI SAO reference spectrum at each across-track position is determined by averaging all spectra collected at that position
between latitudes 30$^\circ$ S and 30$^\circ$ N from the orbit closest in time and with an equatorial crossing closest to 160$^\circ$ W and within
140$^\circ$ W and 180$^\circ$ W (Nowlan et al., 2023). The spectra at the reference locations are also used for slant column reference
sector corrections including HCHO background addition as described below.

The $\Delta$SCD is converted to VCD through Eq. (1).

$\text{VCD} = (\Delta\text{SCD} + \text{SCD}_{\text{Ref}} + \text{SCD}_{\text{B}})/\text{AMF}$ , (1)

Where SCD$_{\text{Ref}}$ is reference sector correction; SCD$_{\text{B}}$ is bias correction; and $\Delta$SCD + SCD$_{\text{Ref}}$ + SCD$_{\text{B}}$ is also referred to as the
corrected slant column. The SCD$_{\text{Ref}}$ corrects the cross-track pixel dependence sensitivity and adds HCHO background slant
columns from the reference region from a chemical transport model (VCD from CTM model$\times$AMF) (Nowlan et al., 2023).
The SCD$_{\text{B}}$ is from the modeled columns of HCHO and used to correct what are primarily latitude-dependent biases in the
retrieved $\Delta$SCD, likely due to interfering absorbers and insufficiently corrected instrument calibration issues (Nowlan et al.,

157  2023).


The AMF defines the mean photon path across the atmosphere and is used in the retrievals to convert slant columns into
vertical columns (Eq. (1)). AMF is calculated by the product of altitude-dependent gas phase HCHO shape factors (S) and
scattering weights (w) integrated along the vertical coordinate (Eq. (2)). Shape factor (S) is the normalized HCHO vertical
number density and calculated from the product of altitude dependent HCHO mixing ratio C and air mass density M
normalized by HCHO column density (see Eq. (3)). The HCHO vertical mixing ratio profile (or *a priori* profile) comes from
a GEOS-Chem 2018 monthly climatology at 0.5$^\circ \times$ 0.5$^\circ$ resolution. Scattering weights are altitude-dependent HCHO





measurement sensitivities and are calculated from a vector multiple-scatter multilayer discrete-ordinate radiative transfer
model (VLIDORT) v2.8 (Spurr, 2006). Scattering weights depend on the viewing angles, surface albedo, surface pressure
and clouds. The scattering and absorption of abnormal aerosol loading can also affect scattering weights and may not be
properly represented in calculated scattering weights (e.g., unpredicted biomass burning plumes).
$$\text{AMF} = \int_0^z w\,(z)S(z)dz \,, \tag{2}$$
$$S(z) = \frac{c(z)M(z)}{\int_0^z C(z)M(z)dz} \,, \tag{3}$$

Previous comparisons of airborne to satellite HCHO data used OMI SAO v003 (Wolfe et al., 2019; Zhu et al., 2020). OMI
SAO v003 retrieves slant column density using direct differential optical absorption spectroscopy (DOAS) (Gonzalez Aabd et
al., 2015). To show the difference between OMI SAO v004 and OMI SAO v003, the global maps of HCHO from OMI SAO
v004, OMI SAO v003 and their difference with the temporal average for the ATom-1 time period are provided in supplementary
Figure S2.

### 177 2.2.2 OMPS-NPP SAO

OMPS is onboard the joint NASA/NOAA Suomi National Polar-orbiting Partnership (NPP) satellite that was launched in 2011
with a spatial resolution at nadir of $50 \times 50$ km and daily global coverage. OMPS also has an equatorial crossing time of about
13:30 local time. The retrieval of OMPS-SAO is described in Nowlan et al., (2023), and is identical to that described above
(Sect. 2.1.1). The spatial and temporal coverage of OMPS and OMI differ due to both their native spatial resolutions and the
OMI row anomaly (González Abad et al., 2016).

### 183 2.2.3. OMI BIRA

OMI BIRA is the European Union Quality Assurance for Essential Climate Variables (QA4ECV) product (De Smedt et al.,
2015; Zara et al., 2018). It is basically the same retrieval algorithm as the operational product of TROPOspheric Monitoring
Instrument (TROPOMI) launched in October 2017 (De Smedt et al., 2021). The detailed retrieval algorithms are described in
De Smedt et al. (2018) and only a brief description is provided here. OMI BIRA retrieval also involves two steps. The
spectra fitting window is 328.5–359 nm, slightly larger than SAO retrievals.
For OMI BIRA, slant column densities are converted to vertical columns as Eq. (5).
$$\text{VCD} = (\Delta\text{SCD} - N_{s,0})/\text{AMF} + N_{v,0} \,, \tag{5}$$
$N_{s,0}$, the slant column correction, corrects the remaining global offset and possible stripes (cross-track pixel dependence
sensitivity) of the differential slant column. $N_{v,0}$, the vertical column correction, is from the TM5 model to compensate for a
background HCHO level due to methane oxidation in the equatorial Pacific (De Smedt et al., 2021). The corrected slant column
is defined as differential slant column ($\Delta$SCD) minus slant column correction ($N_{s,0}$) plus the product of vertical column



correction ($N_{v,0}$) and AMF. The OMI BIRA gas profile comes from TM5-MP model 1º × 1º daily data. The radiative transfer
model for OMI BIRA is VLIDORT v2.7 (De Smedt et al., 2017) , a slightly different version from that used in the SAO
retrievals. In the OMI BIRA retrieval, the location of reference sector for destriping and global offset correction is between
latitudes 5˚S and 5˚N and longitudes 120˚W and 180˚W and for zonal correction is between latitudes 90˚S and 90˚N and
longitudes 120˚W and 180˚W (De Smedt et al., 2017). Considering the locations of the reference sectors (see Figure S3),
understanding of the HCHO concentration over the clean Pacific Ocean is important for evaluating the accuracy of satellite
HCHO retrievals.

Table 1. Parameters in satellite retrievals

| | Nadir pixel resolution (km²) | Fitting windows (nm) | HCHO absorption cross section | Chemical Transport Model (CTM) | Radiative transfer model and wavelength for calculation | Trace gas profiles | Reference sector locations |
|---|---|---|---|---|---|---|---|
| OMI SAO | 24 × 13 | 328.5-356.5 | HITRAN (Chance and Orphal, 2011), 300 K | GEOS-Chem v09-01-03 | VLIDORT v2.8, 340 nm | GEOS-Chem 2018 monthly climatology 0.5º×0.5º | Latitudes :30˚S - 30˚N longitudes: an equatorial crossing closest to 160˚W and between 140˚W and 180˚W |
| OMPS-NPP SAO | 50 × 50 | the same as above | the same as above | the same as above | the same as above | the same as above | the same as above |
| OMI BIRA | 24 × 13 | 328.5-359 | Meller and Moortgat, 2000, 298K | TM5-MP | VLIDORT v2.7, 340 nm | TM5-MP daily profiles, 1º×1º | Destripping and global offset correction: latitudes 5˚S–5˚N, longitudes 120˚W–180˚W; Zonal correction: latitudes 90˚S–90˚N, longitudes 120˚W −180˚W |


### 2.2.4 Retrieval uncertainties

Uncertainties in satellite retrievals come from instrument calibrations, slant column fitting processes, slant column
corrections, and AMF calculations. Averaging damps random uncertainties, while the systematic uncertainties remain
(Nowlan et al., 2023). Instrument noise, choice of fitting windows, HCHO cross-section error, surface reflectance, *a priori*
profiles, vertical distribution and properties of clouds and aerosols all can contribute to the overall systematic uncertainties of
satellite HCHO products. In the OMPS SAO retrieval, the systematic uncertainty in corrected slant column is about 20%
(Nowlan et al., 2023). The error from surface reflectance is about 5% over water, from aerosols is about 0.3% in global mean
(but considerably larger in polluted regions and individual observations), from profile shape is 5% at low HCHO
concentrations, from cloud fraction is 1% and from cloud pressure is 5-15% (Nowlan et al., 2023). The total systematic error
is about 26%. We assume other retrievals have similar or smaller systematic errors, as OMPS SAO uses climatological cloud
pressure and probably has the largest uncertainty (Nowlan et al., 2023).

### 2.2.5 Satellite data filtering and gridding

OMI SAO and OMPS SAO HCHO data use the same categories to filter the data while OMI BIRA use slightly different
filtering categories. SAO L2 data with solar zenith angle > 60°, cloud fraction > 40%, main data quality flag not equal to 0





are excluded. OMI BIRA L2 data with solar zenith angle > 60°, cloud fraction > 40%, and processing error flag ≠ 0 but ≤
255 are excluded.

The 3-D data such as gas profiles are first re-gridded to a universal vertical grid coordinate for all pixels. The L2 2-D and 3-
D data are then gridded into 0.5° × 0.5° using an area weighted average (e.g, AMF, Gas Profiles), shown in Eq. (6), or
uncertainty weighted average (e.g., HCHO column density), as shown in Eq. (7).

$$\overline{C_{ai}} = \frac{\sum_n C_n A_{n,i}}{\sum_n A_{n,i}},$$ (6)
$$\overline{C_i} = \frac{\sum_n \frac{C_n A_{n,i}}{A_n E_n^2}}{\sum_n \frac{A_{n,i}}{A_n E_n^2}},$$ (7)
where is $\overline{C_{ai}}$ is the area weighted average value (such as AMF) for grid $i$, $\overline{C_i}$ is the uncertainty weighted average value (such
as HCHO column density) for grid $i$, $C_n$ is the HCHO column density for pixel n, $A_{n,i}$ is the area contribution of pixel n to
grid $i$, $A_n$ is the total area of pixel n, and $E_n$ is the uncertainty of HCHO column density for pixel n.

The gridded 0.5° × 0.5° daily satellite HCHO data are averaged over each ATom period (ATom-1: 29 July – 23 August,
2016; ATom-2: 26 January – 21 February, 2017; ATom-3: 28 September – 27 October, 2017; ATom-4: 24 April – 21 May,
2018). Differential slant column, slant column corrected, and vertical column all use uncertainty weighted averaging (Eq.
(6)). For comparison to in situ HCHO composite columns, the latitude and longitude coverage of the in situ profile are
identified and the satellite HCHO grids intercepted with the profile latitudes and longitudes are averaged to compare to the
calculated in situ HCHO composite column.
**3. Results and discussions**
**3.1 Global distribution and seasonal variability of HCHO in the marine atmosphere**
Global HCHO distributions from all three retrievals and in situ composite columns across the Pacific and Atlantic Oceans
show enhancement in the tropics and decrease toward polar regions (Figures 1 and 2). The HCHO vertical column density
over the remote ocean atmosphere ranges from about $4 \times 10^{15}$ molecules cm$^{-3}$ at low latitudes to about $1 \times 10^{15}$ molecules cm$^{-3}$
at high latitudes. These large-scale features reflect similar latitudinal and seasonal variability in OH and photolysis rates.
Although the random noise for satellite HCHO such as OMPS SAO is about $3.5 \times 10^{15}$ molecules cm$^{-3}$ (Nowlan et al., 2023),
averaging in time and space largely reduces the noise and thus the variability of HCHO in the remote ocean atmosphere can
be well captured with near one-month average data. In situ HCHO columns corroborate the latitudinal-dependent HCHO
trend over the remote oceans.

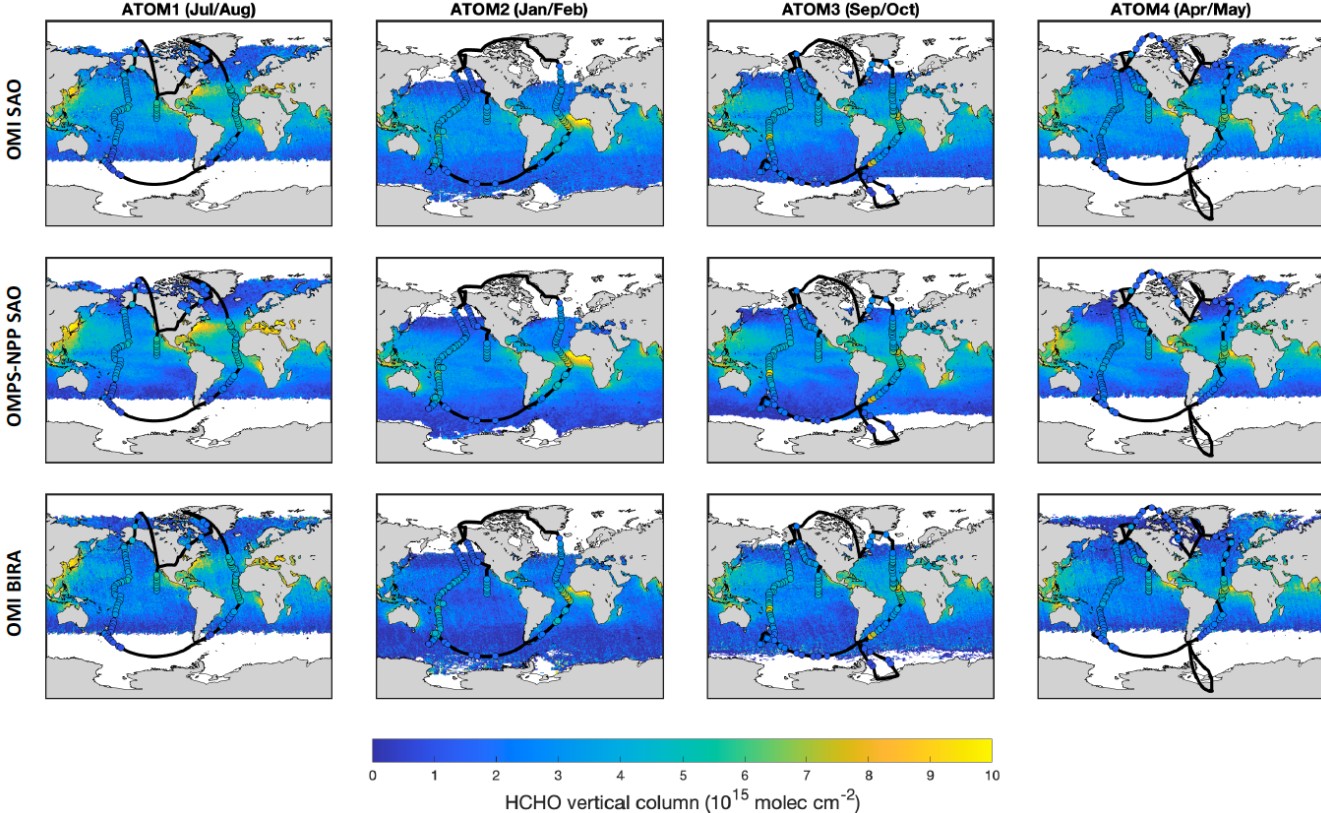

**Figure 1. Maps of HCHO vertical column density from three satellite retrievals (OMI-SAO, OMPS-SAO and OMI-BIRA, top to bottom) over the oceans during four ATom measurement seasons (left to right) overlaid with in situ HCHO columns (colored dots) along the ATom flight tracks (black lines). The color bar for both satellite and in situ HCHO composite columns is the same and saturates at both ends.**

Besides background methane oxidation, continental outflow also affects marine HCHO. All three satellite retrievals capture the continental outflow of HCHO or its precursors from East Asia, North America, Africa, and South Asia (Figure 1). These enhancements can be significant; for example, HCHO off the Atlantic coast of equatorial Africa in February reaches $1.1\times 10^{16}$ molecules cm$^{-2}$, sampled by ATom-2. ATom-3 observed enhanced HCHO in the vicinity of Fiji island when DC8 landed and took off (Figure 1). This enhancement is likely due to local emissions and thus is excluded from the analysis below. Enhanced HCHO mixing ratios near Argentina is also observed during ATom-3. This may be due to a transient biomass burning plume, as black carbon is also enhanced at this time, though carbon monoxide (CO) is not enhanced. Satellite HCHO data also do not show a sustained enhancement at this location. The in situ HCHO composite column enhancement in ATom-3 near Argentina was also excluded from the following analysis.

Zonal mean HCHO varies with season (Figure 2). During ATom-1 in July and August (boreal summer), peak HCHO occurs in a broad band between latitudes near 15-35$^{\circ}$ N. During ATom 2 in January and February (austral summer), the maximum HCHO latitude occurs near 5$^{\circ}$ S with enhancement extending down to 45$^{\circ}$ S. Maximum HCHO latitudes for ATom-3 and -4



Atmospheric
Measurement
Techniques

Discussions

(spring/fall) are near the equator ($\pm5°$). For ATom-3 and ATom-4, HCHO is systematically higher in the Northern
Hemisphere for comparable latitudes (e.g., $3 \times 10^{15}$ molecules cm$^{-2}$ at 50° N vs. $2 \times 10^{15}$ molecules cm$^{-2}$ at 50° S for ATom-
3). This, along with the asymmetric summer maxima, suggests that HCHO precursors (e.g., methane and other VOCs) are
more concentrated in the Northern Hemisphere and impact the distribution of HCHO over the remote ocean. Increased NO$_x$
and ozone can also promote formation of OH and thus HCHO.

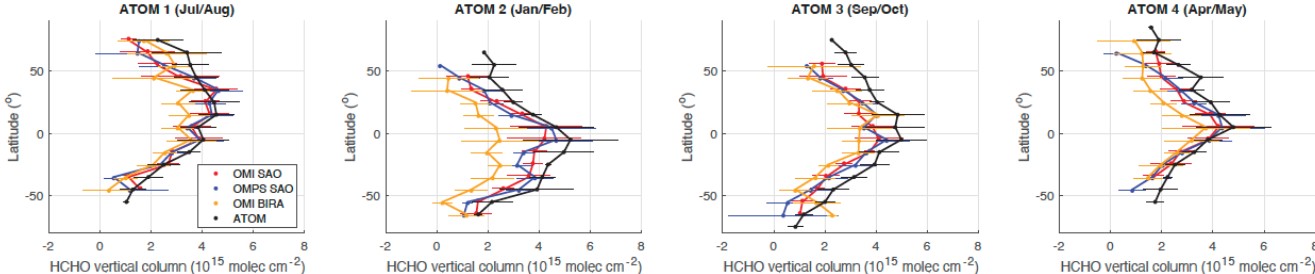

**Figure 2. HCHO column density from three satellite retrievals (OMI SAO in red, OMPS SAO in blue, and OMI BIRA in orange)**
**and ATom in situ measurements (black) at different latitudes. The dots represent the averaged column density for ± 5° latitude**
**bins and the bars are the standard deviation within the latitude bin. OMI SAO error bars are vertically offset for clarity.**
Continental outflows enhance HCHO near the coast, varying with seasons (Figure 1). Enhancements near East Asia, South
Asia, North America and Europe are highest during boreal summertime (ATom-1) and lowest during boreal winter time
(ATom-2), reflecting higher biogenic emissions and stronger photochemistry during the former. Biomass burning outflow
from Africa also varied with seasons, peaking during ATom-2 north of the equator and ATom-1 south of the equator. The
biomass burning outflows from Africa impacted the ATom-2, -3 and -4 flights and thus the Atlantic transits have higher
HCHO concentrations than Pacific transits. The biomass burning impacted air masses are not excluded in the analysis
because the African biomass burning outflows affect large areas and likely happen yearly and can be considered as part of
the background.
**3.2 Comparison between retrievals and in situ HCHO columns**
Comparison of satellite HCHO with ATom in situ composite column densities provides validation of satellite HCHO over
remote oceans, assuming ATom sampling is representative of the monthly average conditions. All retrievals (OMI SAO,
OMPS SAO and OMI BIRA) are well correlated with in situ integrated columns ($r^2 \geq 0.74$), with slopes ranging from 0.75
to 1.33 for individual seasons and negative intercepts on the order of $1 \times 10^{15}$ molecules cm$^{-2}$ (Figure 3; Table 2). The
uncertainty in HCHO above 10 km is on the order of $10^{14}$ molecules cm$^{-2}$ and cannot account for the negative intercepts.
Persistent negative intercepts may suggest a low bias or offset in all retrievals, maybe related to modeled HCHO. GEOS-
Chem predicted HCHO was higher than observed during TRACE-P (Singh et al., 2004) and in-between two HCHO
observations during INTEX-A (Millet et al., 2006). Considering all retrievals, OMI SAO exhibits the best agreement with
ATom overall (slope = $1.02 \pm 0.05$, intercept = $-0.8 \pm 0.2 \times 10^{15}$ molecules cm$^{-2}$). Considering individual ATom
deployments, retrievals fall closest to the 1:1 line against ATom columns for ATom1 (Figure 3). For ATom-2, OMI BIRA

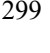



also appears to be systematically low with a slope of 0.75 ± 0.09. Low OMI BIRA HCHO in ATom-2 is also evident in
Figure 2.

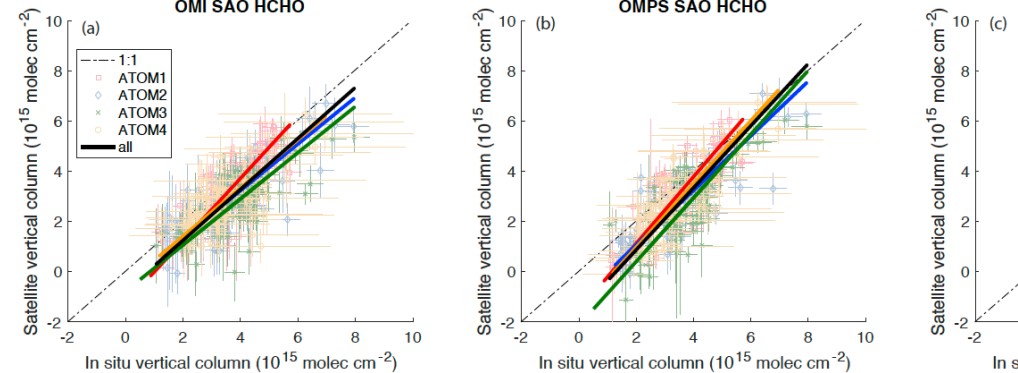

**Figure 3. Scattered plots of satellite HCHO vertical columns from OMI SAO (a), OMPS SAO (b), and OMI BIRA (c) retrievals**
**versus in situ integrated vertical columns from four seasons: ATom-1 (red), ATom-2 (blue), ATom-3 (green) and ATom-4**
**(orange). Error bars for satellite data are the standard deviation of the averaged grid cells, while error bars for in situ composite**
**columns are propagated from the uncertainty of the in situ measurements: $\pm 10\% + 10$ pptv (or $\sim 4.8 \times 10^{14}$ molec cm$^{-2}$) for ISAF**
**and $\pm 40\%$ (or 40 pptv, whichever is greater) (or $\sim 1.9 \times 10^{15}$ molecules cm$^{-2}$) for TOGA. The colored lines and black line are the**
**equally weighted linear regression for each ATom and the total ATom data, respectively. The 1: 1 line is shown as the dashed line.**
**The slopes and intercepts are summarized in Table 2. The higher standard deviations of OMI BIRA HCHO data are due to some**
**large negative values not filtered and do not imply large variation of OMI BIRA HCHO data.**

Table 2 Parameters for linear fits of satellite retrievals against ATom observations (see Figure 3).

|  | OMI SAO | | | OMPS SAO | | | OMI BIRA | | |
|---|---|---|---|---|---|---|---|---|---|
|  | Slope | Intercept (×10$^{15}$) | r$^2$ | Slope | Intercept (×10$^{15}$) | r$^2$ | Slope | Intercept (×10$^{15}$) | r$^2$ |
| ATom-1 | 1.24±0.11 | -1.26±0.41 | 0.84±0.06 | 1.33±0.10 | -1.54±0.39 | 0.85±0.06 | 0.99±0.12 | -0.86±0.45 | 0.77±0.10 |
| ATom-2 | 0.93±0.07 | -0.49±0.27 | 0.85±0.07 | 1.09±0.07 | -1.11±0.24 | 0.89±0.06 | 0.75±0.09 | -1.20±0.31 | 0.78±0.09 |
| ATom-3 | 0.92±0.08 | -0.79±0.33 | 0.81±0.08 | 1.27±0.10 | -2.14±0.39 | 0.83±0.07 | 1.28±0.14 | -2.37±0.54 | 0.77±0.09 |
| ATom-4 | 0.96±0.11 | -0.53±0.38 | 0.79±0.10 | 1.26±0.10 | -1.56±0.34 | 0.85±0.07 | 1.09±0.16 | -1.61±0.55 | 0.74±0.11 |
| all | 1.02±0.05 | -0.79±0.18 | 0.58±0.04 | 1.24±0.05 | -1.61±0.18 | 0.66±0.03 | 1.12±0.07 | -1.84±0.27 | 0.42±0.04 |


The agreement between satellite HCHO retrievals and in situ composite columns is latitude-dependent (Figure 2). Generally,
negative bias is smaller near the equator and more pronounced at higher latitudes, although this depends on season (Figure
2). This is probably indicative of issues with latitude-dependent background corrections in satellite retrievals and/or global
model bias. A more holistic investigation of relevant models with other ATom observations (e.g., ozone, OH, CO, and other
trace gases) may help diagnose the latter. Reactive bromine chemistry at high latitudes may also play a role in the latitude-
dependent satellite retrieval bias as bromine oxide (BrO) is a potential interfering absorber at pptv levels with high
uncertainty in its concentration distribution. Although the difference between in situ composite columns and satellite
retrievals are larger toward high latitudes, in situ composite columns are higher than satellite retrievals even near the equator
during ATom-3 (Figure 2). Satellite OMI SAO and OMPS SAO HCHO vertical columns are closer to OMI BIRA during
ATom-3 than other seasons (Figure 2).



### 3.3 Differences between retrievals

The three satellite HCHO retrievals all captured the patterns of the enhanced continental outflows though there are some small differences among them. Due to the sensor signal to noise ratio and pixel resolution, OMPS SAO HCHO maps are smoother (less noisy) than OMI HCHO data. OMPS SAO HCHO tends to have higher values near continental outflow regions and lower values far away from the outflow regions than OMI SAO HCHO (Figure 1). Although most of the continental outflows are not captured by the ATom flight tracks that were usually over the remote oceans far away from the continents, OMPS SAO HCHO columns along the ATom flight tracks are still higher than OMI SAO at high values and lower than OMI SAO at lower values (Figure 3). OMI BIRA HCHO columns usually have lower values than the other two retrievals, especially for ATom2.

### 3.4 Factors contributing to retrieval differences

Here we compare each component of satellite retrievals that could contribute to the retrieval differences. First, OMI SAO and OMI BIRA HCHO data are compared to probe the impact of different algorithms on retrievals from the same sensor. Second, OMI SAO and OMPS SAO data are examined to investigate the impact of different sensors on the data with the same retrieval algorithm.

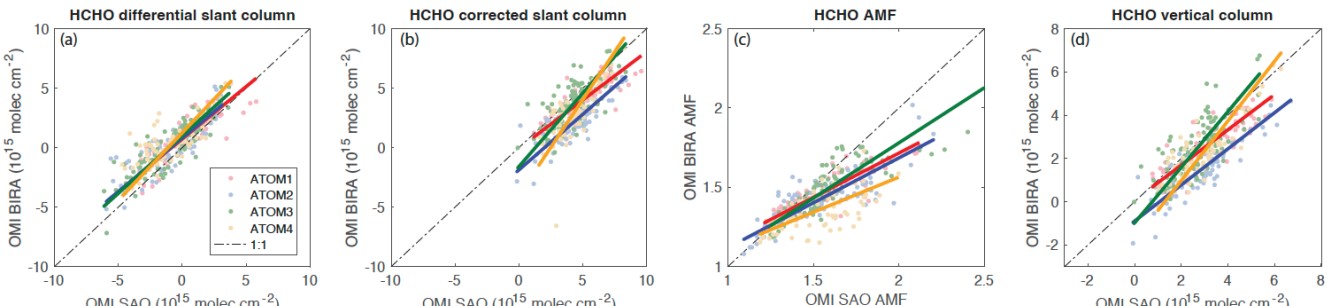

**Figure 4** Comparison of the (a) HCHO differential slant column, (b) corrected slant column, (c) AMF, and (d) vertical column between OMI BIRA and OMI SAO for each ATom deployment.

Table 3 Parameters for linear fits of OMI BIRA vs OMI SAO retrievals subsampled over ATom flights tracks (see Figure 4).

| | OMI BIRA vs OMI SAO | | | | | | | | | | | |
| | Differential slant column | | | Corrected slant column | | | AMF | | | Vertical column | | |
| | Slope | Intercept ($\times 10^{15}$) | $r^2$ | Slope | Intercept ($\times 10^{15}$) | $r^2$ | Slope | Intercept | $r^2$ | Slope | Intercept | $r^2$ |
|---|---|---|---|---|---|---|---|---|---|---|---|---|
| ATom-1 | 0.88±0.06 | 0.73±0.12 | 0.72±0.05 | 0.81±0.07 | -0.06±0.37 | 0.66±0.07 | 0.55±0.03 | 0.60±0.05 | 0.79±0.06 | 0.82±0.06 | 0.06±0.22 | 0.70±0.06 |
| ATom-2 | 0.90±0.07 | 0.80±0.14 | 0.70±0.05 | 0.94±0.08 | -1.87±0.36 | 0.63±0.08 | 0.58±0.05 | 0.55±0.07 | 0.64±0.07 | 0.84±0.06 | -0.90±0.18 | 0.71±0.06 |
| ATom-3 | 0.97±0.06 | 1.00±0.13 | 0.74±0.04 | 1.23±0.13 | -1.59±0.58 | 0.47±0.07 | 0.69±0.38 | 0.39±0.06 | 0.76±0.05 | 1.28±0.11 | -0.98±0.33 | 0.56±0.05 |
| ATom-4 | 1.13±0.14 | 1.25±0.22 | 0.45±0.12 | 1.61±0.16 | -3.99±0.68 | 0.57±0.11 | 0.44±0.06 | 0.68±0.10 | 0.37±0.07 | 1.38±0.13 | -1.79±0.39 | 0.57±0.13 |





### 3.4.1 OMI SAO vs OMI BIRA

Differential HCHO slant column densities of OMI BIRA and OMI SAO are generally well correlated with slopes of 0.8 – 1.1 and intercepts of about $1\times 10^{15}$ molecules cm$^{-2}$ (Figure 4a, Table 3). Because slant column values are the differential between measured spectra over ocean and the reference sector spectrum, the slant column values go both positive and negative. Differences in differential slant columns may be due to both the retrieval wavelength range and the reference spectrum (Table 1). The strong $O_4$ absorption at 356.5–359 nm may contribute to the higher differential HCHO slant column in OMI BIRA than OMI SAO; Nowlan et al. (2023) shows that the difference between the two fitting windows is typically $< 4 \times 10^{14}$ molecules cm$^{-2}$ at clean background levels. HCHO absorption cross sections used in the two retrievals come from different sources (see Table 1). The different chosen reference spectra may also contribute to the difference between OMI BIRA and OMI SAO slant columns. The OMI SAO reference spectrum at each across-track position is the average of spectra between 30° N to 30° S in the orbit with closest in time and an equator crossing closest to 160°W and within 140°–180°W (Nowlan et al., 2023). The OMI BIRA reference spectrum is using the daily average spectrum from the day before for each across-track row in equatorial pacific region (latitude 5° N to 5° S and longitude 120°–180° W) (De Smedt et al., 2018).

Conversion to corrected slant columns generally reduces agreement between the two retrievals (Figure 4b). Background HCHO slant columns at slightly different reference sectors and potential other corrections from different models are added so the corrected slant columns are shifted to mostly positive values. The variability in slopes in the two retrievals among different ATom seasons is larger in corrected slant column than in differential slant column, which may be caused by the differences in background HCHO concentrations from different models results. The background HCHO and corrections for OMI SAO and OMPS SAO are from a GEOS-Chem 2018 monthly climatology (Nowlan et al., 2023), while the background HCHO and corrections for OMI BIRA is from the TM5-MP model daily data (De Smedt et al., 2021, 2017).

Despite the relatively large differences in AMFs, agreement between retrievals for corrected slant columns and vertical columns is relatively similar (Figure 4d). Slopes are similar, and correlation coefficients actually improve by 5-10% with the vertical columns. Partly this is because the range of variability in AMFs is small (factor of 2) compared to variability in corrected slant columns (factor of 10). This implies that systematic uncertainties in AMFs are likely minor contributors to overall retrieval error in remote environments.





### 3.4.2 OMI SAO vs OMPS SAO

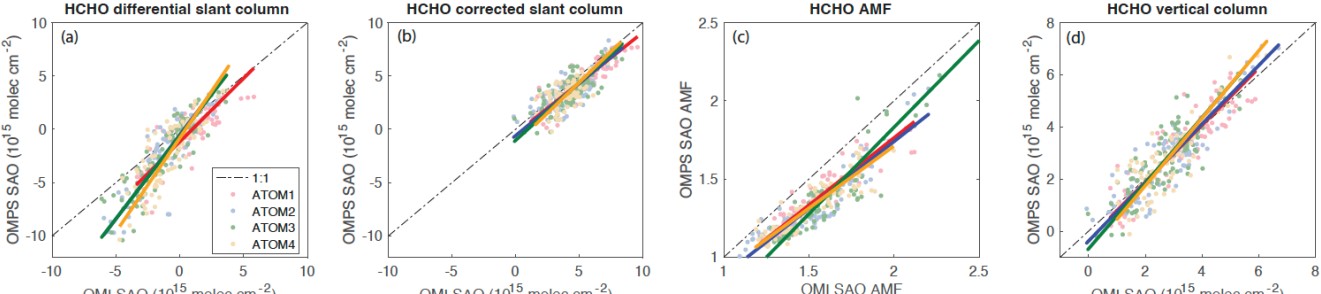

**Figure 5. Comparison of the (a) HCHO differential slant column, (b) corrected slant column, (c) AMF, and (d) vertical column between OMPS SAO and OMI SAO for each ATom deployment.**

Table 4. Parameters for linear fits of OMPS SAO vs OMI SAO retrievals subsampled over ATom flights tracks (see Figure 5).

| | OMI SAO vs OMPS SAO | | | | | | | | | | | |
|---|---|---|---|---|---|---|---|---|---|---|---|---|
| | Differential slant column | | | Corrected slant column | | | AMF | | | Vertical column | | |
| | Slope | Intercept (×10^15) | r^2 | Slope | Intercept (×10^15) | r^2 | Slope | Intercept | r^2 | Slope | Intercept | r^2 |
| ATom-1 | 1.19±0.10 | -1.17±0.18 | 0.65±0.06 | 0.95±0.07 | -0.41±0.35 | 0.74±0.05 | 0.86±0.04 | 0.04±0.06 | 0.85±0.02 | 1.09±0.07 | -0.31±0.25 | 0.77±0.04 |
| ATom-2 | 1.58±0.10 | -0.52±0.20 | 0.77±0.06 | 0.98±0.08 | -0.60±0.37 | 0.63±0.07 | 0.86±0.04 | 0.03±0.06 | 0.84±0.03 | 1.12±0.06 | -0.38±0.19 | 0.80±0.05 |
| ATom-3 | 1.55±0.08 | -0.62±0.17 | 0.81±0.02 | 1.08±0.09 | -1.04±0.39 | 0.61±0.06 | 1.11±0.04 | -0.39±0.07 | 0.86±0.04 | 1.26±0.08 | -0.68±0.23 | 0.72±0.05 |
| ATom-4 | 1.76±0.13 | -0.82±0.23 | 0.69±0.05 | 1.15±0.10 | -1.37±0.45 | 0.61±0.08 | 0.80±0.05 | 0.12±0.07 | 0.80±0.03 | 1.30±0.08 | -0.85±0.24 | 0.77±0.05 |

Differential slant columns from OMI SAO and OMPS SAO are generally well correlated ($r^2$ = 0.65–0.81), with OMPS SAO slant columns lower at low values (Figure 5a). Different sensor properties and calibrations for the two sensors are likely explanations for these differences. Correction for cross-track pixel dependence sensitivity, HCHO background slant column, and latitude-dependent biases greatly improves agreement, with slopes near 1 for corrected slant columns (Figure 5b).

The AMF of OMPS SAO is usually lower than OMI SAO (Figure 5c). Because the *a priori* gas profiles and scattering weights for OMPS SAO and OMI SAO with the same retrieval algorithms are from the same models, their AMF difference could be due to the different pixel size and the related cloud product, with OMPS SAO using climatology cloud pressure (Nowlan et al., 2023) in scattering weight calculation. The low OMPS SAO to OMI SAO AMF ratios brought the ratios of their vertical columns slightly higher than the ratios of their corrected slant columns. The correlation between OMPS SAO and OMI SAO is improved after normalization by AMF to yield vertical columns, which is similar to the comparison of OMI SAO and OMI BIRA, but the slopes get slightly further from 1.

Although uncertainties in AMFs are likely minor contributors to overall retrieval error in remote ocean environments, roles of *a priori* profiles and scattering weights in contributing to the differences in AMF among the three retrievals are explored. Shape factors (S), scattering weights (SW), AMF density (S×SW×10^6), and AMF accumulative density function for season average are shown in Figure 6. To better visualize the profiles, shape factors only below 15 km are shown in Figure 6, although ATom shape factors are available in altitudes up to ~10 km and satellite shape factors are up to 40 km. The





average shape factors of OMI SAO and OMPS SAO are identical due to the same chemical transport model outputs GEOS-
Chem 2018 monthly climatology 0.5º × 0.5º data used. OMI BIRA shape factors are close to SAO shape factors except for
ATom-2, where OMI BIRA has higher HCHO values near the surface. To be noted, OMI BIRA HCHO is significantly
lower than the other two retrievals during ATom-2 (Figure 2). ATom shape factors tend to have lower distribution near the
surface than satellite shape factors. OMI SAO and OMPS SAO scattering weights come from the same radiative transfer
model VLIDORT v2.8 while scattering weights of OMI BIRA come from VLIDORT v2.7. However, OMPS SAO uses a
different cloud product for the scattering weights calculation. The climatology cloud data OMPS SAO uses are fixed at the
same height all the time for a given location, giving OMPS SAO the characteristic bump feature near 2 km and leading to the
difference in AMF density distribution with OMI SAO and OMI BIRA having one peak along altitude axis at ~ 3 km and
OMPS SAO having a peak at higher altitude (~ 4 km). AMF density distribution profiles using ATom *a priori* profiles show
similar maximum altitudes to the OMI satellite data. Due to the order of calculations, AMFs estimated from average *a priori*
and scattering weight of OMI BIRA are not always smaller than that of OMI SAO as shown in Figure 4c. Three satellite
retrievals all show that about 10% of AMF density distribution is above 10 km, which was not measured by ATom
observations.



Figure 6. Air mass factor (AMF) components shape factor (S) (a-d), scattering weights (SW) (e-h), and the product of S and SW (S× SW) defined as AMF density (i-l) and the AMF cumulative density function (m-p) for the three satellite retrievals (red: OMI-SAO, blue: OMPS-NPP SAO, orange: OMI BIRA, black: derived from ATom measurements) and four seasons (different columns). ATom shape factor S comes from ATom in situ profiles.

## 4. Conclusions

We use in situ HCHO measurements from four seasonal deployments of the NASA ATom airborne mission to evaluate three satellite retrievals (OMI-SAO (v004), OMPS-NPP SAO, and OMI-BIRA) of total HCHO columns. All retrievals correlate with in situ composite columns over the remote marine regions, with OMI-SAO retrieval exhibiting the best agreement. Retrievals also capture the patterns of zonal gradients and seasonal variability, with the best agreement near the equator and





persistent negative bias at higher latitudes. OMI BIRA HCHO is consistently lower than the other two retrievals, with
anomalously low HCHO in February 2017. The discovery of latitude-dependent biases provides useful information for future
improvement of satellite HCHO retrievals.

Intercomparison of results from intermediate retrieval steps reveals the influence of different algorithms and different
sensors on derived HCHO columns. Notably, 1) OMI BIRA and SAO differences seem to be mainly due to the applied
background corrections, 2) OMI and OMPS have different differential SCDs but corrections fix most of that though OMPS
is still slightly higher at high values and lower at low values than OMI, and 3) AMFs can be significantly different, but they
don't seem to affect agreement between retrievals because the dynamic range of AMFs is relatively small.

Evaluation of retrievals using in situ composite columns implies that 1) retrievals of HCHO in remote regions do contain
actual measurement information, but models also affect retrieval accuracy; 2) retrievals may be sufficient as inputs to
parameterize OH or other species not directly measured from space, but the potential latitude-dependent systematic bias of
up to $2 \times 10^{15}$ molecules cm$^{-2}$, which is substantial in the remote marine regions, should be considered; 3) this study
considered one species in a relatively simple region of the atmosphere, and retrieval differences will vary by molecule and
by location. Vertical profiles from in situ instruments are clearly crucial for providing ground truth needed to validate
satellite retrievals.
**Data availability**
The NASA ATom data are available at DAAC archive (https://doi.org/10.3334/ORNLDAAC/1925). OMI SAO v004 data
are available at Harvard SAO server (https://waps.cfa.harvard.edu/sao_atmos/data/omi_hcho/). OMPS SAO data are
available at NASA GES DISC archive (https://doi.org/10.5067/IIM1GHT07QA8). The OMI BIRA data are available at
temis Website (https://www.temis.nl/qa4ecv/hcho.html; https://doi.org/10.18758/71021031).

**Author contributions**
GMW initiated and guided the project. AEK searched for the best satellite datasets to use, contacted satellite people to get
the satellite dataset, and used codes from JL to process some satellite data. JL wrote codes to grid and process the satellite
datasets and used codes from GMW to calculate in situ composite column. JL re-processed and analyzed the data and
discussed the results with GMW and JN. JL wrote the manuscript. GMW, JMSC, and TFH collected ATom ISAF data.
GGA, CRN, ZA and IDS provided satellite data. GGA provided the key equation to grid the satellite data. CRN provided





additional useful information for the satellite retrievals. ECA and RSH collected ATom TOGA data. All authors reviewed
and/or commented on the manuscript.
**Competing interests**
At least one of the (co-)authors is a member of the editorial board of Atmospheric Measurement Techniques.
**Acknowledgments**
JL, GMW, AEK, JN, JMSC, and TFH are supported by NASA Tropospheric Composition Program (TCP). JL, AEK, JN,
and JMSC are also supported by NOAA Atmospheric Chemistry, Carbon Cycle and Climate (AC4) program
(NA19OAR4310164). GGA, CRN and ZA are supported by NASA Making Earth System Data Records for Use in Research
Environments (80NSSC18M0091), algorithm maintenance for SAO standard OMI products (80NSSC21K0177), and
Algorithm maintenance for SAO OMI products (80NSSC24K0120). GGA and CRN are also supported by NASA Science of
Terra, Aqua, and Suomi-NPP (80NSSC18K0691). ECA and RSH are supported by the NSF National Center for
Atmospheric Research, which is a major facility sponsored by the U.S. National Science Foundation under Cooperative
Agreement No. 1852977.

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
