# Peer review of "Validation of formaldehyde products from three satellite retrievals (OMI SAO,"

_Atmospheric Measurement Techniques, 2024_

## Referee Comment (RC1)

**Review comments on "Validation of formaldehyde products from three satellite retrievals (OMI SAO, OMPS-NPP SAO, and OMI BIRA) in the marine atmosphere with four seasons of ATom aircraft observations"**

This manuscript systematically analyzes the differences and sources of remote sensing datasets of formaldehyde column concentrations over the oceans using ATom data and multiple satellite HCHO inversion results. I believe that this work has important implications for both satellite dataset developers and users, especially given the scarcity of validation of oceanic atmospheric observations. The article should be finally published after addressing the issues below.

Major comments:

1. On the significance of the study for data developers and users The oceanic atmosphere HCHO retrieval may be highly noisy due to the instrument detection limits. Therefore, this study is of great importance to both satellite data developers and users in this area. In my opinion, quantitative assessment of the data quality and futher suggestions on retrieval improvement should be emphasized in the manuscript (e.g., abstract and introduction) in relation to the existing knowledge and shortcomings in the data application work, in order to directly highlight the significance and conclusions of the study to the readers. For example, it would be informative for readers to have the mean bias for each satellite HCHO products in the abstract and conclusion section.

2. Regarding the heterogeneity and transformation of atom and satellite observations The transformation of atom in situ observations into atmospheric column concentrations is essential to the comparisons results described in this paper. Although partially mentioned in L120-130, some doubts may remain. For example, missing atom data and the absence of observations in the upper atmosphere (> 10km) require interpolation and averaging, how much do these treatments affect the results? What percentage of Atom data is missing? Are there any uncertainties in the molecule number concentration method? Also in L127-129, "Average gas profiles from OMI SAO HCHO retrievals are used to estimate the contribution of HCHO above 10 km to the total HCHO column": how to derive the ratio of HCHO columns above 10 km from OMI SAO retrievals ? It should be total column HCHO retrieved from OMI spectral measurements. Does such conversion relying on OMI SAO HCHO affects the comparisons with other satellite products such as BIRA product.

3. When comparing different satellite products, may the author use the convolution of averaging kernels in satellite HCHO rertievals with Atom measurements, to minimizing the impact the using different a priori profiles in AMF calculations.

Minor comments:

1. L243-245: the unit of column density should be molecules cm -2?
2. Table 2-4: other metrics such as mean bias should be added and discussed in the main text.

---

## Author Comment (AC1)

Responses to reviewers' comments on "Validation of formaldehyde products from three satellite retrievals (OMI SAO, OMPS-NPP SAO, and OMI BIRA) in the marine atmosphere with four seasons of ATom aircraft observations"

We appreciate the valuable feedback and support from two reviewers and Jean-Francois Muller regarding the publication of this manuscript following revisions. In response to their suggestions, we have carefully revised the manuscript. To facilitate the review process, we have included the reviewers' comments in black text, with our responses in blue. All comments have been addressed, and the corresponding changes to the manuscript are tracked.

Referee #2

This manuscript systematically analyzes the differences and sources of remote sensing datasets of formaldehyde column concentrations over the oceans using ATom data and multiple satellite HCHO inversion results. I believe that this work has important implications for both satellite dataset developers and users, especially given the scarcity of validation of oceanic atmospheric observations. The article should be finally published after addressing the issues below.

Major comments:

1. On the significance of the study for data developers and users: The oceanic atmosphere HCHO retrieval may be highly noisy due to the instrument detection limits. Therefore, this study is of great importance to both satellite data developers and users in this area. In my opinion, quantitative assessment of the data quality and futher suggestions on retrieval improvement should be emphasized in the manuscript (e.g., abstract and introduction) in relation to the existing knowledge and shortcomings in the data application work, in order to directly highlight the significance and conclusions of the study to the readers. For example, it would be informative for readers to have the mean bias for each satellite HCHO products in the abstract and conclusion section.

   Mean biases are added to the abstract, results and discussion and conclusion. Abstract: added "The agreement is also reflected in the mean bias (MB) for OMI SAO $(-0.73\pm0.87)\times10^{15}$ molec cm$^{-2}$, OMPS SAO $(-0.76\pm0.88)\times10^{15}$ molec cm$^{-2}$, and OMI BIRA $(-1.40\pm1.11)\times10^{15}$ molec cm$^{-2}$."

   Conclusion: added "The mean bias for OMI SAO, OMPS SAO, and OMI BIRA is -0.73 $(\pm0.87)\times10^{15}$ molec cm$^{-2}$, -0.76 $(\pm0.88)\times10^{15}$ molec cm$^{-2}$, and -1.40 $(\pm1.11)\times10^{15}$ molec cm$^{-2}$, respectively."

The mean bias values are also added to Table 2, 3, and 4 and discussed in the paper.

Introduction lines 89-97 describe the potential HCHO retrieval issues over the remote ocean atmosphere. Changed "Consequently, validation of satellite HCHO over the remote ocean would aid in assessing the satellite's ability to capture background HCHO levels accurately and enhancing our understanding of these baseline levels.  To "Consequently, quantitative assessment of satellite HCHO over the remote ocean is crucial for assessing the satellite's ability to accurately capture background HCHO levels and deepening our understanding of these baseline levels." Added "Refining satellite HCHO retrievals will reduce potential bias in applications such as estimating VOC emissions and atmospheric oxidant levels."

2. Regarding the heterogeneity and transformation of atom and satellite observations: The transformation of atom in situ observations into atmospheric column concentrations is essential to the comparisons results described in this paper. Although partially mentioned in L120-130, some doubts may remain. For example, missing atom data and the absence of observations in the upper atmosphere (> 10km) require interpolation and averaging, how much do these treatments affect the results? What percentage of Atom data is missing? Are there any uncertainties in the molecule number concentration method? Also in L127-129, "Average gas profiles from OMI SAO HCHO retrievals are used to estimate the contribution of HCHO above 10 km to the total HCHO column": how to derive the ratio of HCHO columns above 10 km from OMI SAO retrievals? It should be total column HCHO retrieved from OMI spectral measurements. Does such conversion relying on OMI SAO HCHO affects the comparisons with other satellite products such as BIRA product.

We have revised the text to better explain our process for selecting columns, including considerations of missing data percentage. For the portion of HCHO above 10 km, we rely on model results (satellite a priori profiles) and we have provided a clearer explanation of this process. Additionally, details on how molecule number concentration is calculated have been added to the Supplementary information.

Changed "Columns are filtered to include only profiles with solar zenith angle smaller than 80°, minimum altitude <= 600 m, maximum altitude >= 8 km, fraction of missing interpolated grids < 0.2, and fraction of missing extrapolated data <0.25." to " Columns are filtered to include only profiles with solar zenith angle smaller than 80°, minimum altitude <= 600 m, maximum altitude >= 8 km, fraction of missing measured data in the altitude profiles < 0.2, and fraction of missing extrapolated data between 0 to 10 km <0.25. The average missing interpolated data within 0 – 10 km is 8%, mostly due to lower resolution TOGA data are used during ATOM 4. The

data gaps are typically small and lack significant structure, so we expect them to contribute to random error rather than introduce any systematic bias. The average missing extrapolated data between 0 – 10 km is 5%. "

Changed "Average gas profiles from OMI SAO HCHO retrievals are used to estimate the contribution of HCHO above 10 km to the total HCHO column. " to "Most HCHO > 10 km were not measured during ATom field campaign so modeled results, average gas profiles from OMI SAO HCHO retrievals,  are used to estimate the contribution of HCHO above 10 km to the total HCHO column. The gas profiles in OMI SAO retrieval are from GEOS-Chem 2018 monthly climatology 0.5º×0.5º (Table 1)."

Line 128 changed "The calculated fraction of HCHO above 10 km (relative to the total column) is 0.045± 0.002." to "The fraction of HCHO above 10 km (relative to the total column) is 0.045± 0.002, calculated by the integrated gas profiles above 10 km divided by the integrated gas profiles from 0- 40 km."

SI Added "Molecule number concentration is calculated by Eq.(S1)

$$M = Na \times P/R/T \qquad (S1)$$

Where Na is Avogadro's number $6.022 \times 10^{23}$ mol-1; P is pressure in mbar; R is gas constant $8.314 \times 10^4$ cm$^3$ mbar K$^{-1}$ mol$^{-1}$ and T is temperature in K.

3. When comparing different satellite products, may the author use the convolution of averaging kernels in satellite HCHO rertievals with Atom measurements, to minimizing the impact the using different a priori profiles in AMF calculations.

Line 400 added "The convolution of averaging kernels in satellite HCHO retrievals with ATom measurements was not performed for three reasons: 1) AMFs are likely minor contributors to overall retrieval error in the study regions. 2) In the remote oceanic atmosphere, the shape factors for three retrievals are generally very similar (Figure 6a). Adjusting them to match ATom measurements could systematically alter the AMF of the retrievals but it would not significantly affect the differences among them. 3) HCHO level distributions or shape factors above 10 km are not available from ATom measurements, potentially introducing additional uncertainties in the clean oceanic atmosphere due to high scattering weights (or averaging kernels) at high altitudes."

Minor comments:
1. L243-245: the unit of column density should be molecules cm -2?

The units are corrected.

2. Table 2-4: other metrics such as mean bias should be added and discussed in the main text
   Mean biases are added in Table 2-4. They are discussed in the main text.

---

## Author Comment (AC2)

Responses to reviewers' comments on "Validation of formaldehyde products from three satellite retrievals (OMI SAO, OMPS-NPP SAO, and OMI BIRA) in the marine atmosphere with four seasons of ATom aircraft observations"

We appreciate the valuable feedback and support from two reviewers and Jean-Francois Muller regarding the publication of this manuscript following revisions. In response to their suggestions, we have carefully revised the manuscript. To facilitate the review process, we have included the reviewers' comments in black text, with our responses in blue. All comments have been addressed, and the corresponding changes to the manuscript are tracked.

Referee #3:

Liao et al., utilize a four-season deployment of Atom aircraft observations to validate three HCHO retrieval products. They demonstrate that these HCHO products generally capture the spatial and seasonal distribution of HCHO in the remote ocean-atmosphere albeit with a low bias. An important result of this study is that the biases in slant column corrections have larger impacts on retrieval than AMFs. The paper is well-organized and includes technical details that fit well into the scope of AMT. I hope the authors can address the following comments before the paper is accepted for publication in AMT.

Major comments:

The conclusion of this paper could be further strengthened. While the study effectively validates these HCHO retrievals and addresses differences in HCHO columns across latitudes and seasons, it would be valuable for the authors to provide practical advice to users of these products. For example, do the authors have any recommendations on which retrieval product is preferable? Would averaging across multiple products yield more accurate results than using a single product? Alternatively, should the spread among the three products be treated as an indicator of uncertainty in HCHO retrieval?

According to the analysis of this study, we recommend OMI-SAO (v004) at least for the remote ocean atmosphere studies because this retrieval has the best agreement and smallest mean biases compared to ATom in situ data.

In the abstract, we have added "All retrievals are correlated with ATom integrated columns over remote oceans, with OMI SAO (v004) showing the best agreement." We now added "This is also reflected in the mean bias (MB) for OMI SAO (-0.73±0.87), OMPS SAO (-0.76±0.88), and OMI BIRA (-1.40±1.11). We recommend the OMI-SAO (v004) retrieval for remote ocean atmosphere studies."

We would not advise averaging retrievals or using the spread as a measure of uncertainty, as the ATom profiles should serve as ground truth.

Other comments:

1. The OMI satellite overpass time is 1:30 pm local time while the Atom observations were conducted throughout the day. How do you account for the time difference when comparing HCHO retrievals to Atom observations?

   Section 3.2 added "Data on the diurnal variation of HCHO columns in the remote oceanic atmospheric are very limited (e.g., the Mauna Loa site in the supplementary information of Vigouroux et al. (2018)). Given the possible diurnal variation of HCHO, the difference between aircraft sampling time and satellite overpass time (1:30 pm) may account for some, but not the majority, of the discrepancies between satellite and ATom measurements at high latitudes (Fig. 4S and 5S). The differences across latitudes due to time variation may amount to approximately $0.2 \times 10^{15}$ molecules cm$^{-2}$, based on the simulation results (Fig. 4S and 5S). Further research is needed to more accurately quantify the diurnal variation of HCHO over oceanic regions."

   Supplementary material added:

[Figure]

**Figure S4. (a) Map of ATom1 flight track color-coded with local time. ATom 2, 3, and 4 maps are similar to ATom1 and not shown here. (b) Diurnal variation of HCHO columns with maximum value of $2.0 \times 10^{15}$ molec cm$^{-2}$ at 1: 00 pm is simulated, as an example. The diurnal variation is based on the profiles from Bruno Franco et al. (2016) and the maximum value is selected based on average satellite HCHO measurements at the northern high latitudes. It is important to note that the diurnal variation shown in (b) likely represents the upper limit of diurnal HCHO column fluctuations in the remote oceanic atmosphere, especially in high latitudes, as suggested by the measurements reported in Vigouroux et al. (2018).**

[Figure]

**Figure 5S. The latitude-averaged distribution of simulated HCHO columns, using HCHO columns as a function of local time (Figure S4b) and the local time of the ATom flight tracks. This figure highlights the differences between satellite and ATom measurements across latitudes, which arise solely from the time discrepancies-- 1:30 pm local time for satellite measurements and varying local times for ATom measurements (Figure S4a). When comparing these measurements across latitudes, ATom measurements may appear higher than satellite measurements at higher latitudes (e.g., ATom1 70⁰ N compared to 30⁰ N in Figure 5S) due to local time differences. However, the local time effect contributing about 0.2 × 10¹⁵ molec cm⁻², is relatively minor compared to the overall differences between satellite and ATom measurements across latitudes (e.g., Figure 2 ATom1 70⁰ N vs. 30⁰ N). The relatively large variation in high southern latitudes may suggest that the simulated HCHO column variability is significantly overestimated for this region.**

2. Line 121: the ascents and descents of aircraft measurement cover 200-450 km in horizontal distance, which is larger than the pixel size of satellite retrievals. Also, the aircraft provides *in-situ* measurements while the satellite measures pixel by pixel. How do you account for the differences in the spatial scales of these two observations?

   Line 121-122 change "In situ HCHO columns are calculated using the method described in Wolfe et al. (2019)" to "In situ HCHO columns are compared to the average of satellite grid cells intersected by the in situ profile area and calculated using the method described in Wolfe et al. (2019)."

3. It is unclear how you treat cloudy conditions when mapping satellite retrievals to ATOM observations. Do you only select satellite/ATOM observation under clear sky conditions?

   In section 2.2.5 Line 219- 221 we stated "SAO L2 data with solar zenith angle > 60°, cloud fraction > 40%, main data quality flag not equal to 0 are excluded. OMI BIRA L2 data with solar zenith angle > 60°, cloud fraction > 40%, and processing error flag ≠ 0 but ≤ 255 are excluded."

4. Figure 2: why is there an enhancement of the HCHO column at ~ -60 latitude bins in the OMI BIRA retrieval products?

   Section 3.2 added "The enhancement of HCHO columns around the -60⁰ latitude bins may be attributed to noise in the OMI BIRA retrievals, specifically anomalous elevated values around filtering gaps when zoomed in, as observed over high southern latitudes in ATom 2 and ATom 3 (Figure 1)."

5. Line 313: since negative bias is more pronounced at higher latitudes, does it suggest that the latitude-dependent background correction is insufficient?

   In Line 314, we stated "This is probably indicative of issues with latitude-dependent background corrections in satellite retrievals and/or global model bias."

6. Line 367-368: what does "variability" refer to here? If it refers to uncertainties, a factor of 10 seems too large. If it refers to the full range of corrected slant columns, I don't understand why this implies that uncertainties in AMF are a minor contributor to overall retrieval error

   Changed "Partly this is because the range of variability in AMFs is small (factor of 2) compared to variability in corrected slant columns (factor of 10)." To " This is primarily because the low OMI BIRA to OMI SAO AMF ratios correspond to the low HCHO column values and the data are spread."

---

## Author Comment (AC3)

Responses to reviewers' comments on "Validation of formaldehyde products from three satellite retrievals (OMI SAO, OMPS-NPP SAO, and OMI BIRA) in the marine atmosphere with four seasons of ATom aircraft observations"

We appreciate the valuable feedback and support from two reviewers and Jean-Francois Muller regarding the publication of this manuscript following revisions. In response to their suggestions, we have carefully revised the manuscript. To facilitate the review process, we have included the reviewers' comments in black text, with our responses in blue. All comments have been addressed, and the corresponding changes to the manuscript are tracked.

Comments from Jean-Francois Muller

Hello,

Athough I appreciate the very nice work presented in this paper, I am concerned by the spatial averaging of the OMI column data, as shown on Equation 7. Why this uncertainty weighting? Higher columns have generally a higher uncertainty, in absolute terms (their RELATIVE uncertainty is however generally lower). Equation 7 gives therefore less weight to higher columns. As a consequence, the average is too low. The authors should repeat their calculations by using a regular weighting as given by Equation 6. I played with the OMI data myself and found that the averaging has a substantial impact on the results. I am very curious to see the impact on the analysis presented in this paper.

Best regards,

Jean-Francois Muller

There is no clear consensus within the satellite community on whether uncertainty weighting is preferable.

Sect 3.2 added "Uncertainty-weighted satellite HCHO columns (Eq. (6), all figures in main text) are generally slightly lower than area-weighted satellite HCHO columns (Eq. (7), Figure S6) over the remote oceanic atmosphere, particularly in the OMI BIRA retrieval. However, the different weighting methods do not affect the overall conclusions of the analysis results."

SI added Figure S6

[Figure]

Figure 6S. Area weighted HCHO column density from three satellite retrievals (OMI SAO in red, OMPS SAO in blue, and OMI BIRA in orange) and ATom in situ measurements (black) at different latitudes. The dots represent the averaged column density for ± 5° latitude bins and the bars are the standard deviation within the latitude bin. OMI SAO error bars are vertically offset for clarity.